

# Relation semantic fusion in subgraph for inductive link prediction in knowledge graphs

Hongbo Liu[1], Jicang Lu[1], Tianzhi Zhang[1], Xuemei Hou[1] and Peng An[2]

[1] School of Data and Target Engineering, Information Engineering University, ZhengZhou, Henan, China
[2] School of Cyberspace Security, Zhengzhou University, Zhengzhou, Henan, China

## ABSTRACT

Inductive link prediction (ILP) in knowledge graphs (KGs) aims to predict missing links between entities that were not seen during the training phase. Recent some subgraph-based methods have shown some advancements, but they all overlook the relational semantics between entities during subgraph extraction. To overcome this limitation, we introduce a novel inductive link prediction model named SASILP (Structure and Semantic Inductive Link Prediction), which comprehensively incorporates relational semantics in both subgraph extraction and node initialization processes. The model employs a random walk strategy to calculate the structural scores of neighboring nodes and utilizes an enhanced graph attention network to determine their semantic scores. By integrating both structural and semantic scores, SASILP strategically selects key nodes to form a subgraph. Furthermore, the subgraph is initialized with a node initialization technique that integrates information about neighboring relations. The experiments conducted on benchmark datasets demonstrate that SASILP outperforms state-of-the-art methods on inductive link prediction tasks, and verify the effectiveness of our approach.

## INTRODUCTION

A knowledge graph is a collection of triplets organized in the structure of *(h,r,t)*. It plays a critical role in domains such as information retrieval (*Wang et al., 2020*), intelligent recommendation (*Yu et al., 2020*), and intelligent Q & A (*Kaiser, Saha Roy & Weikum, 2021*). However, most existing knowledge graphs are incomplete. To address this issue, numerous link prediction methods have been developed. Most of these methods use an embedding-based approach, which involves mapping entities and relations into a vector space and obtaining low-dimensional representations through training. A scoring function is then used to evaluate the plausibility of a triplet's existence. This approach, known as transductive inference (*Ji et al., 2021*), can be used to infer missing entities or relations, as shown in Figs. 1A and 1B.

Corresponding author
Hongbo Liu, lhb921@163.com

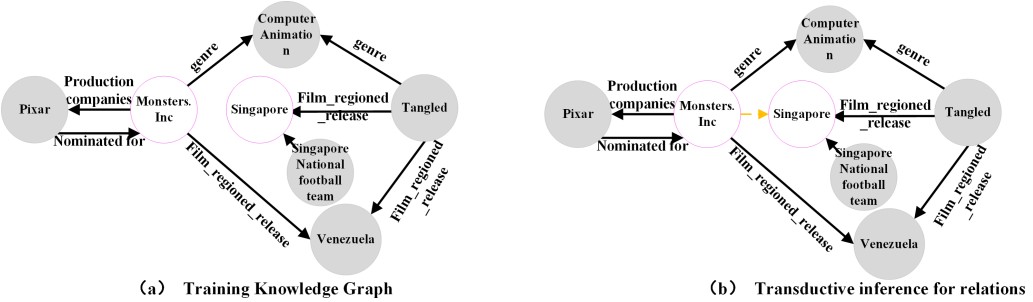

**Figure 1** (A–B) Transductive link prediction in knowledge graph.

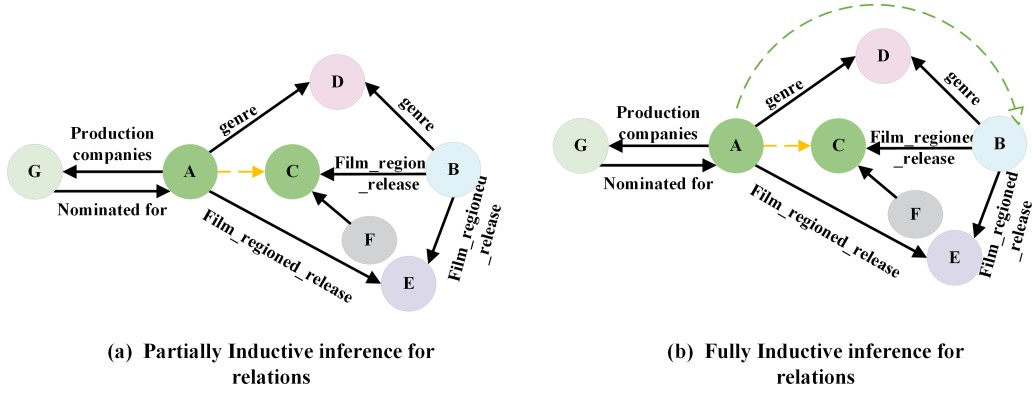

**Figure 2** (A–B) Inductive link prediction in knowledge graph.

However, embedding-based approaches encounter a significant challenge in practice: their effectiveness relies on the assumption that all entity representations have been acquired. If the test set includes entities that are absent from the training set, the model must be retrained. In real-world business scenarios, new entities and relations continuously emerge, necessitating the model's ability to adapt to the dynamically changing data environment. Frequent model retraining can be resource-intensive and time-consuming, making it impractical for real business deployments. To address the link prediction problem, which involves new entities and relations in the test set, researchers have proposed inductive link prediction methods. These methods can be classified into two forms: Fully Inductive and Partially Inductive. In the fully inductive method, the entities observed during training and testing do not overlap, while relations at inference time consist of a mixture of known and new relations, as shown in Fig. 2B. Conversely, the partially inductive approach involves testing entities that did not occur in the training, while relations at inference time remain consistent with that observed during training, as shown in Fig. 2A.

The inductive link prediction discussed in our paper refers to partial inductive link prediction methods. These methods typically involve three main steps: subgraph extraction, node initialization representation, and graph neural network aggregation.

Subgraph extraction methods can be divided into two categories. The first category of methods uses the k-hop distance between a node and the head or tail entity related to the target relation as a measure of node importance. According to this perspective, nodes that are nearer to the head or tail entity linked to the target relation are considered more important. Therefore, when extracting a subgraph, the method first identifies and obtains the set of k-hop neighboring entities of the head or tail entity associated with the target relation. Subsequently, the subgraph is constructed by calculating the intersection of these two sets.

The second category of methods uses probabilistic, such as PageRank (PR) (*Brin & Page, 2012*), Hyperlink-Induced Topic Search (HITS) (*Kleinberg, 1999*), and Personalized PageRank (PPR) (*Lofgren & Goel, 2013*), to evaluate the importance of nodes through random walk. These methods construct subgraphs based on node scores, which reflect the importance of nodes in the knowledge graph with respect to the target relation.

In knowledge graphs, relations in triplets are rich in semantic information, and entities with the same relation are semantically similar. For instance, in the triplet (X, film_regional_released, Y), based on the relation, we can infer that the head entity (X) represents the movie, and the tail entity (Y) represents the place name. This relational semantics enables us to capture semantic associations between entities more effectively.

Subgraph extraction methods based on random walk and k-hop node sets fail to adequately capture the complex dependencies between nodes. The attention mechanism can be used to learn the edge weights in the graph, which reflect the interdependence between nodes.

Inspired by the previous work, we propose a novel inductive link prediction model called Structure and Semantic Inductive Link Prediction (SASILP). This model can predict missing links between unseen nodes. SASILP employs a method that combines graph structure and relational semantic information to extract subgraphs and initialize subgraph representations. Specifically, we use graph attention networks and random walk algorithms to obtain the nodes' semantic and structural scores, respectively, and use these scores as the basis for subgraph extraction. Moreover, we aggregate entity-related relations to initialize entities. In this way, SASILP can effectively incorporate relational information into the enclosing subgraph, thus improving the performance of inductive link prediction.

The main contributions of this paper are as follows:

- We introduce a novel subgraph extraction method for inductive link prediction, which realizes subgraph extraction by combining structural and relational semantic information.
- We propose a novel method for node initialization that considers the underlying relation semantics present in all known triplets. This approach can provide comprehensive information to represent emerging entities and meet the requirements of the induction scenario.
- We conducted inductive link prediction experiments on the benchmark datasets. The results show that SASILP outperforms the baseline models in 3 out of 4 metrics on WN18RR_v2, WN18RR_v3 and FB15K237_v4.

# RELATED WORK

## Path and rule-based link prediction

These approaches use path information or logical rules between entities in a knowledge graph to predict links. The Path Ranking Algorithm (PRA) (*Lao & Cohen, 2010*) analyses the path patterns between entities. AMIE (*Galárraga et al., 2013*), RuleN (*Meilicke et al., 2018*), Neural-Lp (*Wang et al., 2019*) and DRUM (*Yang, Yang & Cohen, 2017*) learn logical rules from the knowledge graph. AMIE and RuleN generate logical rules based on path search traversal, which has limited scalability. Neural-Lp generates logical rules through recurrent neural networks, while DRUM learns rules in a differentiable manner. These rules can capture semantic information that is independent of entities. However, they still face challenges in handling complex logic and large scale data, as well as improving the accuracy and interpretability of rules.

## Embedding-based link prediction

Embedding-based link prediction predicts links by mapping entities and relations in the knowledge graph to embeddings in a low-dimensional space. The embeddings capture semantic information about the entities and relations, enabling the computation of possible relations between entities. TransE (*Bordes et al., 2013*) is one of the earliest models for embedding knowledge graphs, which assumes that the sum of the embeddings of the head and tail entities should be close to the relation embeddings. TransE is a model suitable for 1-to-1 relations. However, it has limitations when dealing with 1-to-many, many-to-1, and many-to-many relations. To handle complex relation types better, TransH (*Wang et al., 2014*) TransR (*Lin et al., 2015*), and TransD (*Ji et al., 2015*) are proposed. DistMult (*Yang et al., 2015*) and ComplEx (*Trouillon et al., 2016*) capture complex interactions between entities by using different scoring functions. ComplEx introduces complex embeddings to model symmetric and antisymmetric relations. With the development of neural network technology, neural network-based knowledge graph representation models have shown good performance in representation learning, M-DCN (*Zhang et al., 2020*) utilize multi-scale processing to optimize quality in the convolution layer for knowledge graph embedding. It generates various sizes of convolution filters to learn different characteristics between the embeddings of entity and relation. SHGNet (*Li et al., 2023*) designs a hierarchical aggregation architecture for feature propagation. And the method incorporates neighbor information *via* node aggregation and relation weighting, which can selectively aggregate informative features. TEGS (*Li et al., 2024*) amalgamates textual encoding and graph structure information, facilitating the concurrent acquisition of contextualized and structured knowledge. These approaches assume that all entities and relations are known during training. They cannot predict entities and relations that have not appeared in the training set. For knowledge graphs where new nodes and edges are constantly being added, embedding-based link prediction methods require frequent re-training to adapt to the changes.

### GNN-based link prediction

In recent years, graph neural networks (GNNs) (*Scarselli et al., 2008*) have made significant progress in link prediction tasks. GNNs efficiently capture both local and global information in the graph structure. Graph Convolutional Network (GCN) (*Kipf & Welling, 2016*) learns the representation of nodes by performing convolutional operations on graphs. Graph attention network (GAT) (*Velickovic et al., 2017*) introduces an attention mechanism that allows the model to dynamically focus on important neighboring nodes. The relational graph convolutional network (R-GCN) (*Schlichtkrull et al., 2018*) is specifically designed to deal with heterogeneous relations in graphs. These methods enhance the expressive power of the model and improve the accuracy of link prediction, however, they cannot predict relations between newly added entities.

To achieve efficient generalization of patterns and predict relations between new entities without frequent model retraining, innovative approaches have been developed. These approaches combine subgraph extraction techniques with GNNs for inductive link prediction. GraiL (*Teru, Denis & Hamilton, 2020*) extracts an enclosing subgraph of target triplet from knowledge graph, and models the subgraph with GNN to capture the topological structure. CoMPILE (*Mai et al., 2021*) highlights the edge direction in the enclosing subgraph based on GraiL. TACT (*Chen et al., 2021*) considers the semantic correlation between two relations in the knowledge graph based on subgraph structural information. SNRI (*Xu et al., 2022*) contains the neighborhood path information of all nodes in the enclosing subgraph. RMPI (*Geng et al., 2023*) first transforms a triplet's surrounding subgraph in the original KG into a new relation view graph, where inter-relation features are more straightforwardly represented. It then learns the embedding of an unseen relation from the relational subgraph by the relational message passing network. LCILP (*Mohamed et al., 2023*) extracts subgraph using a PPR-based local clustering technique.

Although these methods have achieved good results, they only use node information in the graph when selecting nodes for subgraph extraction, without considering the influence of edges, which are important components of knowledge graph.

## METHODS

This section describes the knowledge graph inductive link prediction model entitled SASILP. The objective of the model is to assign a score to the given triplet $(u, r_t, v)$, where u and v represent nodes that were not observed in the training graph but are present in the inference graph, and $r_t$ denotes the target relation between nodes u and *v*. The overall framework of SASILP is shown in Fig. 3, it consists of four parts: importance score calculation, subgraph extraction, node initialization, and score function construction.

### Subgraph extraction

Most of the current inductive link prediction methods rely on the subgraph around the target relation, where the nodes of the subgraph are selected based on their proximity to the target nodes. Inspired by GENI (*Park et al., 2019*), we introduce a novel subgraph extraction method that constructs the subgraph by scoring nodes based on a combination

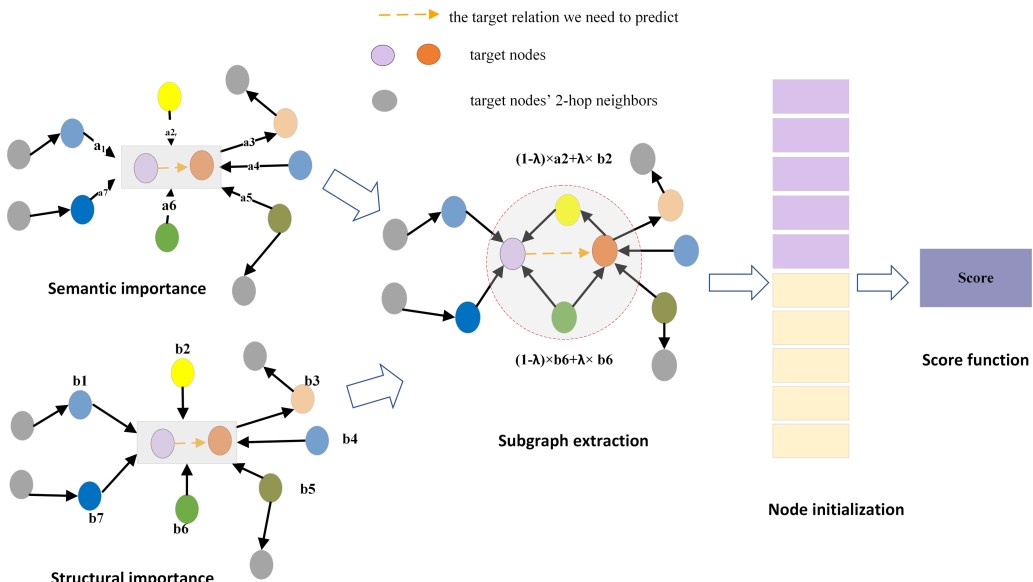

**Figure 3** **The overall framework of SASILP.** a2 denotes the relation semantic score of the yellow node, b2 denotes its structural score. $(1 - \lambda) \times a2 + \lambda \times b2$ represents the final score for it. The red dashed circle represent the subgraph extracted for the target relation.

of node distances and relational semantics. The subgraph extraction method is composed of four parts: (1) Calculation of the semantic importance score of nodes using attention mechanisms. (2) Computation of the structural importance scores of nodes using the PPR algorithm. (3) Calculating the final importance score of nodes by weighting and summing their structural and semantic importance scores. $\lambda$ is a weight paremeter that adjusts the proportion of structural and semantic information in the calculation of node importance. (4) Selecting nodes to form a subgraph based on their scores. Specifically, nodes associated with the target relation are initially ranked in descending order based on their importance scores. Subsequently, these nodes are incrementally added to an initially empty set of seed nodes. After each node addition, the conductance of the current set of seed nodes is computed. The process stops when the calculated conductance value falls below a predefined threshold. The resulting seed node set and its corresponding local subgraph is the subgraph we need.

1. Semantic importance score for node

In knowledge graphs, relations contain crucial semantic information essential for understanding the graph structure and conducting effective node importance assessment. Different semantic relations exert varying influences on node importance. For the target relation 'lives in' in Fig. 4, the relations 'parent of', 'work in', and 'located in' are less important than 'be good at', 'is friend of', and 'husband of'. Nodes connected to more relevant relations should receive higher priority in importance assessment.

GATs are effective in capturing the relative importance between nodes, so we adopt the attention weight as the semantic score of the node. However, GAT may overlook

---

**Algorithm 1** Subgraph extracting algorithm.

---

**Require:** graph G = (V, E) with vertices V and edges E; seed set S= {m, n}; teleportation
probability $\alpha \in (0, 1]$; residual error $\beta$; weight paramater $\lambda$; the embedding of nodes
h; conductance threshold $\phi$;

**Ensure:** subgraph S

1: **for** v in G **do**:
2:     $score_{stru}$ =APPR(G,v, $\alpha$, $\beta$);
3:     $score_{seman}$ =semantic_weight(G,h);
4:     score = $\lambda \times score_{stru}$ +(1-$\lambda$)$\times score_{seman}$;
5: **end for**
6: sort nodes in descending order according to their scores
7: compute the conductance of the set S;
8: **while** conductance $< \phi$ **do**:
9:     add node to the set of seed nodes S;
10: **end while**
11: **return** S;

---

the relational semantic information when dealing with knowledge graphs. To address this
issue, we integrates relational information into the computation of attention weights by
sharing relation embeddings, as shown in Fig. 5. At each layer of the model, node features
are passed to the next layer through the output of the previous layer, while edge features
remain shared across all layers. The node updates its features by considering not only its
information but also the relational information around it. This approach allows for a more
accurate assessment of the importance of nodes in the knowledge graph In graph attention
calculation, the weight coefficient $e_{ij}$ from node $v_i$ to node $v_j$ is defined as

$$e_{ij} = a[s(i) \parallel \sum_{m=1}^{k} p_{ij}^m \parallel s(j)] \tag{1}$$

where $s(i)$ is the embedding of node $i$ obtained by other ways such as TransE or one-hot.
Since there can be many different types of relations between two entities in the knowledge
graph, in order to distinguish different types of edges, $k$ is the number of edge between
node $i$ and $j$, $p_{ij}^m$ represents the embedding of the m-th edge. $a$ is an attention calculation
function. All the computed weight coefficients are normalized by the softmax function:

$$\alpha_{ij} = \frac{\exp(LeakyRelu(e_{ij}))}{\sum_{k \in N_i} \exp(LeakyRelu(e_{ik}))}, e_i \in N_i \tag{2}$$

where $LeakyRelu$ is the activation function. $N_i$ is the set of nodes in the graph.

    2. Structural importance score based on the PPR algorithm.

    The PPR algorithm is primarily used to determine the structural importance score of
entities in the knowledge graph. It sets the head and tail entities of the target relation as the
seed node set, then performs a random walk with a restart from the seed node set, it returns
to the start node with probability c and jumps to neighbor nodes with probability *1-c*. After
many iterations, the algorithm reaches a stable value. The probability distribution can be

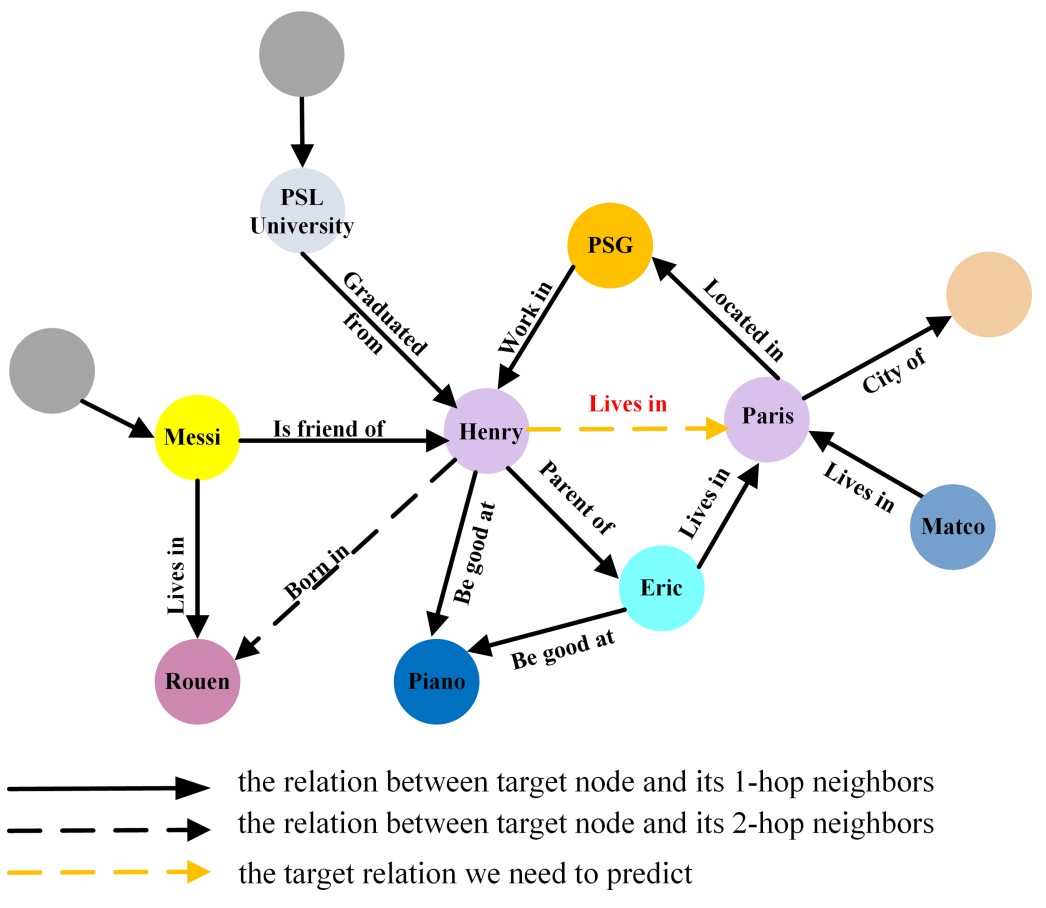

the relation between target node and its 1-hop neighbors

the relation between target node and its 2-hop neighbors

the target relation we need to predict

**Figure 4** **The different importance of different relation for link prediction in knowledge graph.**

viewed as the structural score $score_{stru}$ link to the other nodes of the seed node:

$$\mathbf{r} = cW + (1-c)\mathbf{e} \tag{3}$$

where $c \in (0, 1]$, $W$ denotes the transfer probability matrix, $W[i,j]$ represents the probability from node $j$ to node $i$. $\mathbf{e}$ denotes the initialization vector, If the start node is $i$, then $e\,[i]$ equals 1. Otherwise, $e\,[i]$ equals 0. The vector $\mathbf{r}$ is a column vector that represents the probability of being at node $i$. To reduce computational complexity, an approximate PPR algorithm is commonly used.

3. Node importance score

The final importance score of nodes is obtained by weighting and summing the structural importance score and semantic importance score. The score is defined as follows:

$$score = \lambda \times score_{stru} + (1-\lambda) \times score_{seman} \tag{4}$$

where $\lambda$ denotes the weight given to the structural importance score, and $(1-\lambda)$ denotes the weight given to the semantic importance score.

The entities should be sorted in descending order based on their score and added to the seed set S. Then, the conductance of the set S should be calculated. If the conductance

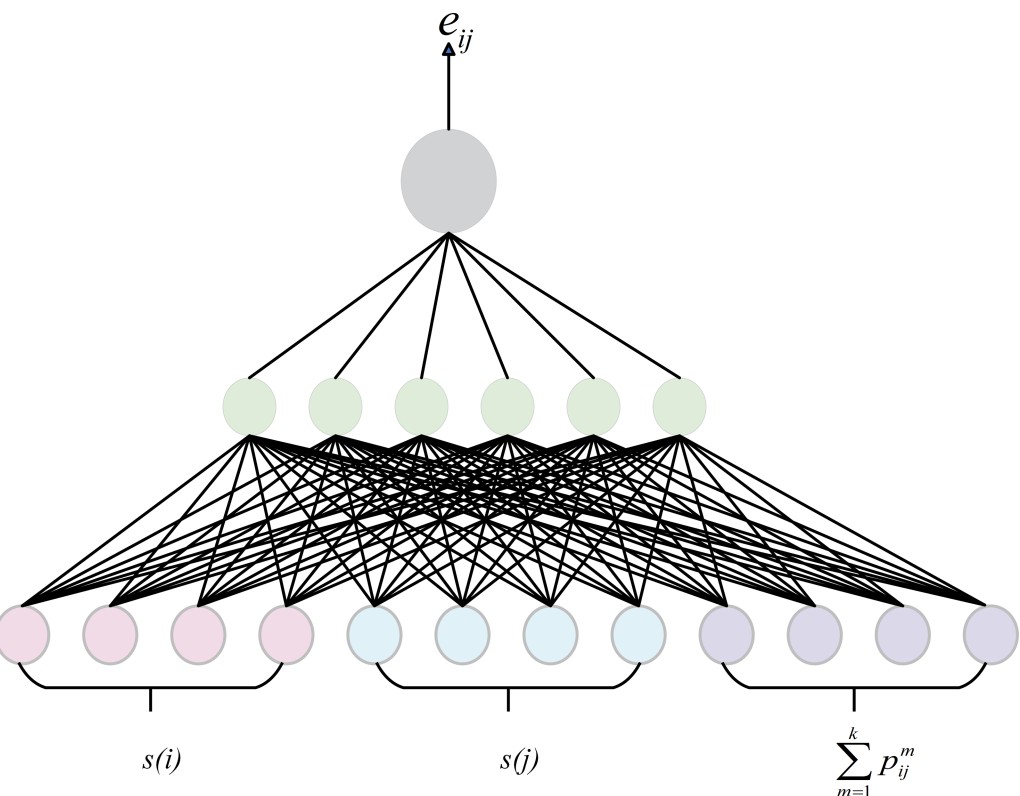

**Figure 5** **Attentation mechanism.**

reaches the specified threshold, the set can be used as an extracted subgraph of the target relation. The calculation of the conductance is defined as:

$$\phi(S) = \frac{|\{(i,j) \in E, i \in S, j \notin S\}|}{\min(vol(S), 2m - vol(S))} \tag{5}$$

where $E$ is the set of edges, $i$ is a point in set $S$, $j$ is a point outside of set $S$, and $vol(S)$ is the sum of the degrees of all points in $S$, $m$ is the number of edges in graph, $2m\text{-}vol(S)$ denotes the sum of the degrees of the nodes not included in $S$.

## Initialization representation of nodes

An appropriate initialization can deploy the intrinsic properties and structural information of entities, and it can bolster the model's representational capacity and generalizability. Most inductive link prediction models for subgraph node initialization adopt structure-based or attribute-based initialization. However, these methods fail to take into account the impact of relations in initializing the embeddings.

In a triplet, entities with the same relation are often semantically similar. Consequently, the relational is instrumental in capturing the essence of both the head and tail entities within the representational framework. By incorporating relation information associated with the entity into the entities themselves, it can implicitly indicate the specific category of the entity at the semantic level, thereby providing more useful information for the

representation of new entities. Based on the aforementioned concept, when initializing the node embedding, the relations associated with the entity are aggregated to form the semantic information embedding of the entity's relation. For entity e, there are both relations to and from it. To clarify the direction of the relation, we use $r_t$ to represent the relation embedding pointing to the entity, and $r_h$ to represent the relation embedding apart from the entity. Since an entity may have multiple relations with other entities, the semantic embedding of entity $e$ is formed by aggregating all the relation embeddings around it, it is defined as

$$e_r = \frac{1}{N_r}(\sum_{i=0}^{n_1} r_h{}^i + \sum_{j=0}^{n_2} r_t{}^j) \quad n_1 + n_2 = N_r \tag{6}$$

where $n_1$ denotes the number of edges apart from entity $e$, $n_2$ denotes the number of edges pointing to entity $e$, and $N_r$ denotes the total number of edges connected to entity $e$. $r_h{}^i$ denotes the relation embedding originating from entity $e$, and $r_t{}^i$ denotes the relation embedding directed from entity $e$.

The representation of entity $e$ is composed of structural information embedding and semantic information embedding $e_r$.

$$e = e_s \parallel e_r \tag{7}$$

the structural information embedding can be obtained by DRNL (*Zhang & Chen, 2018*) in GraiL.

## Subgraph representation

The subgraph representation module uses RGCN to learn the representations of entities in subgraphs, and it is defined as:

$$h_i^{(k+1)} = \sigma \left( \sum_{r \in R} \sum_{v_j \in Nv_i^{(r)}} \frac{1}{a_{i,r}} W_r^{(k)} h_j^{(k)} + W_o^{(k)} h_i^{(k)} \right) \tag{8}$$

where $R$ denotes the set of all relations in the extracted subgraph, and $Nv_i^{(r)}$ denotes the set of neighbors that have relation $r$ with node $v_i$. $a_{i,r}$ is used to perform the normalization operation. $W_r$ is the transformation matrix over relation $r$ in the k-th layer. $w_o$ is the weight parameter corresponding to the node itself, $\sigma()$ is the activation function, and $h_i^{(k)}$ represents the embedding of node $h_i$ in the k-th layer. The pooled average of all nodes in the subgraph is used as the subgraph representation $h_s$:

$$h_s^K = \frac{1}{|V_s|} \sum_{i \in V_s} h_i^K \tag{9}$$

where $V_s$ is the set of node, $h_i$ denotes the embedding of node $i$.

## Scoring and loss function

To obtain the likelihood score of the triplets, the scoring function is designed as follows:

$$f(h, r, t) = [E_h^K \oplus E_t^K \oplus E_r \oplus E_g^K] W \tag{10}$$

where $E_h^K, E_t^K, E_g^K$ are the embeddings of the head entity, tail entity and subgraph in the K-th layer, $E_r$ is the representation of the relation, and $W$ is the weight matrix, and the loss is computed by summing up the scores of all triplets in knowledge graph:

$$Loss = \sum_{(h_i,r_k,t_i) \in G}^{|N|} \max(0, f(h_i, r_k, t_i) - f(\overline{h_i}, r_k, \overline{t_i}) + \gamma) \tag{11}$$

where |N| denotes the number of triplets in knowledge graph, $G$ denotes the knowledge graph, $\gamma$ is margin hyperparameter. $f(h_i, r_k, t_i)$ is the score of positive triplets, and $f(\overline{h_i}, r_k, \overline{t_i})$ is the score of negative triplets. In this paper, the positive triplets are the triplets existing in the knowledge graph, and the negative triplets are constructed by replacing the head (or tail) of the triplets with uniformly sampled random entities.

# EXPERIMENTS

## DataSet

FB15K is a subset of the FreeBase Knowledge Base (KB) (*Bollacker et al., 2008*), which contains general facts about the world with various relation types. FB15K237 is a subset of FB15K, created by removing reversible relational data. The dataset contains 15k subject terms and a total of 237 relations. WN18 is a dataset derived from WordNet KB (*Miller, 1995*), which is a comprehensive database of English vocabulary used to capture lexical relations, such as hyper-subordination between words. WN18 contains a significant number of reversible relations, but WN18RR removes these reversible relations to improve the accuracy of the link prediction task. WN18RR only encompasses 11 relations. To evaluate the model's link prediction capability on a new graph, we adopts GraiL's dataset partitioning method. Four entities are randomly selected as root nodes from the dataset, and the k-hop neighborhood nodes around each root node are merged to create the training dataset. The sample training graph is then removed from the entire graph, and the same method is applied to the remaining graph to obtain the test graph. The process parameters are adjusted to generate a series of graphs with increasing size. The dataset is divided into four parts, denoted as v1, v2, v3, and v4, based on the increasing number. The same method is used to obtain the training and test sets for WN18RR. The dataset statistics are presented in Table 1.

### Baselines and implementation details

1. Baselines

We selected rule-based link prediction methods, namely Neural_LP (*Wang et al., 2019*), DRUM (*Yang, Yang & Cohen, 2017*), and RuleN (*Meilicke et al., 2018*), as well as subgraph extraction-based models, including GraiL (*Teru, Denis & Hamilton, 2020*), SNRI (*Xu et al., 2022*), LCILP (*Mohamed et al., 2023*) and RMPI (*Geng et al., 2023*) to serve as our baseline models. Neural_LP, DRUM: Neural_LP and DRUM both are end-to-end differentiable approaches to learn first-order logical rules from knowledge graph.

RuleN: RuleN learn specific kinds of rules from knowledge graph with a sampling strategy.
GraiL: GraiL extracts enclosing subgraphs around the target relation and models the subgraph with RGCN to predict relations.

**Table 1** **Statics of the dataset.** #entities, #relations, #triplets denote the number of entities, relation and triplets.

| | | WN18RR | | | FB15K237 | | |
|---|---|---|---|---|---|---|---|
| | | #entities | #relations | #triplets | #entities | #relations | #triplets |
| v1 | train | 2,746 | 9 | 6,678 | 2,000 | 183 | 5,226 |
| | test | 922 | 9 | 1,991 | 1,500 | 146 | 2,404 |
| v2 | train | 6,954 | 10 | 18,968 | 3,000 | 203 | 12,085 |
| | test | 2,923 | 10 | 4,863 | 2,000 | 176 | 5,092 |
| v3 | train | 12,078 | 11 | 32,150 | 4,000 | 218 | 22,394 |
| | test | 5,084 | 11 | 7,470 | 3,000 | 187 | 9,137 |
| v4 | train | 3,861 | 9 | 9,842 | 5,000 | 222 | 33,916 |
| | test | 7,208 | 9 | 15,157 | 3,500 | 204 | 14,554 |

SNRI: SNRI integrates complete neighboring relations into the enclosing subgraph and apply MI maximization to inductive link prediction.

LCILP: LCILP extracts subgraph by PPR-based local clustering technique.

RMPI: RMPI transforms a triplet's surrounding subgraph into a new relation graph, and learn the embedding of an unseen relation from the relation graph.

2. Parameter setting

The Python implementation of the experimental code was completed on a server configured with Ubuntu 16.04LTS operating system, featuring Intel(R) Xeon(R) Gold 6348 CPU @ 2.60 GHz configuration, 38GB RAM, and a single v100s GPU. During caculation of random walk, we set the teleportation probability to a commonly used value of $\alpha = 0.15$ (*Vattani, Chakrabarti & Gurevich, 2011*; *Leskovec & Faloutsos, 2006*) for all the datasets, and the approximation parameter $\beta$ to 1e−3 for WN18RR and v1 of FB15K237, while we set it to 1e−4 for the remaining partitions of FB15K237 (v1, v2, v3). During GAT, we adopt MultiLabelSoftMarginLoss as the loss function during training, and set epoch as 50 for WN18RR, 2000 for FB15K237. The model was trained using Pytorch (*Paszke et al., 2019*) and optimized with the Adam (*Kingma & Ba, 2014*) optimizer, with an initial learning rate of 0.001, batch sizes of 8, 16 and 32, hops of 1, 3 and 5, and a training epoch of 50.

### Evaluation metrics

The performance of the model in the knowledge graph is evaluated using Mean Reciprocal Ranking (MRR) and hit rate (hits@@k). MRR averages the inverse of the rankings of all the triplets in the test set. The specific calculation method is as follows:

$$MRR = \frac{1}{|S|} \sum_{i=1}^{|S|} \frac{1}{rank_i} \qquad (12)$$

where $S$ is the set of triplets, $|S|$ represents the number of triplets and $rank_i$ is the predicted rank of links in the first triplet. Hits@k refers to the probability of hitting in the first k results.

$$hits@n = \frac{1}{|S|} \sum_{i=1}^{|S|} I(rank_i \leq n) \qquad (13)$$

where $S$, $|S|$, $rank_i$ and $MRR$ in Eq. (13) involves the same symbols as those used in Eq. (12), and I() denotes the indicator function (which returns a value of 1 if the condition is true, and 0 otherwise). Typically, $n$ is assigned a value of 1, 3, 5, or 10.

## Results and discussion

To compare the performance of SASILP with existing inductive link prediction models, we conducted link prediction experiments on the benchmark datasets WN18RR and FB15K237. We referenced the publicly reported experimental results of the baseline models Neural-LP, DRUM, RuleN and GraiL. However, due to incomplete information in the literature, the results for the SNRI, LCILP and RMPI models are obtained through local execution on our machine.

- Results

Several observations can be obtained from Tables 2 and 3

(1) SASILP outperforms the rule_based models including Neural-LP, DRUM, and RuleN in all metrics. Compared to the best-performing RuleN model, our method has achieved significant improvements across various metrics; specifically, on dataset WN18RR_v3, MRR has been enhanced by 7.27%, and Hits@10 has been improved by 13.88%. Similarly, on FB15K237_v2 dataset, our approach has demonstrated outstanding performance, with MRR showing an improvement of 1.94% and Hits@10 by 4.72%. This is likely due to SASILP model is not only learn path-based rules, but exploit complex patterns in knowledge graph.

(2) Our model performs better in most cases than the latest subgraph_based model including Grail, SNRI, LCILP and RMPI. Compared with the best-performing model LCILP, our method has also shown improvements, specifically, on dataset WN18RR_v3 MRR has been enhanced by 0.77%, and Hits@5 has been improved by 2.4%. It indicates that incorporating relational semantics into subgraph extraction has a beneficial influence on the inductive link prediction experiments.

(3) On FB15K237, RuleN outperforms GraiL on all datasets for Hits@1. This is likely due to RuleN focus on the existence of the rule-based path (*i.e.,* 1 or 0) rather than calculating the probability of the path. However, for Hits@10, GraiL outperforms RuleN, suggesting that GraiL performs better for a wider range of link predictions.

The performance improvements substantiate the effectiveness of our model in the task of inductive link prediction.

- Performance comparison of different subgraph extraction strategies

We conduct additional experiments on dataset WN18RR to analyze the impact of hop selection in extracting subgraphs. The values of MRR, Hits@1, Hits@5, and Hits @@10 were taken for $k = 1$, 3 and 5, respectively. The results are shown in Fig. 6.

Figure 6 shows that the performance in terms of MRR, Hits@1, Hits@5, and Hits@10 is the least favourable on all datasets when $k = 1$. However, there is not much difference when $k = 3$ and $k = 5$. This may be due to the fact that the subgraph consisting of 1-hop neighbors includes a small number of nodes, which limits the availability of contextual semantic information for relation prediction. When $k = 3$, subgraphs can provide sufficient contextual semantic information to improve link prediction performance. However, the

**Table 2  Inductive link prediction performance on WN18RR.**

| Datasets | Model | MRR | Hits@1 | Hits@5 | Hits@10 |
|----------|-------|-----|--------|--------|---------|
| WN18RR_v1 | Neural-Lp | 71.74 | 68.34 | 74.37 | 74.37 |
| | DRUM | 72.46 | 69.60 | 74.37 | 74.37 |
| | RuleN | 79.15 | 76.06 | 81.91 | 80.85 |
| | GraiL | 80.45 | **78.19** | 82.45 | 82.45 |
| | SNRI | 78.99 | 73.14 | 85.11 | **89.63** |
| | LCILP | 79.58 | 73.67 | 87.23 | 88.3 |
| | RMPI | 78.86 | 75.21 | 82.39 | 82.45 |
| | SASILP | **80.49** | 74.73 | **87.77** | 88.56 |
| WN18RR_v2 | Neural-Lp | 68.54 | 66.89 | 68.93 | 68.93 |
| | DRUM | 68.82 | 67.46 | 68.93 | 68.93 |
| | RuleN | 77.82 | 76.53 | 78.23 | 78.23 |
| | GraiL | 78.13 | 76.30 | 78.68 | 78.68 |
| | SNRI | 79.96 | 76.19 | 82.43 | **85.03** |
| | LCILP | 80.86 | 77.78 | 83.22 | 83.22 |
| | RMPI | 78.01 | 76.51 | 78.68 | 78.68 |
| | SASILP | **81.21** | **78.00** | **84.13** | 84.13 |
| WN18RR_v3 | Neural-Lp | 44.23 | 41.16 | 45.92 | 46.18 |
| | DRUM | 44.96 | 42.17 | 46.05 | 46.18 |
| | RuleN | 51.53 | 48.60 | 53.22 | 53.39 |
| | GraiL | 54.11 | 50.33 | 57.19 | 58.43 |
| | SNRI | 46.34 | 41.40 | 47.85 | 53.88 |
| | LCILP | 58.03 | 51.74 | 64.46 | 65.54 |
| | RMPI | 56.22 | **52.89** | 58.31 | 58.84 |
| | SASILP | **58.80** | 51.90 | **66.86** | **67.27** |
| WN18RR_v4 | Neural-Lp | 67.14 | 65.84 | 67.13 | 67.13 |
| | DRUM | 67.27 | 66.11 | 67.13 | 67.13 |
| | RuleN | 71.65 | 70.57 | 71.59 | 71.59 |
| | GraiL | 73.84 | 72.39 | 73.41 | 73.41 |
| | SNRI | 73.41 | 67.74 | 78.31 | **81.32** |
| | LCILP | **77.88** | **75.72** | 79.43 | 79.43 |
| | RMPI | 73.43 | 72.3 | 73.41 | 73.41 |
| | SASILP | 77.31 | 74.46 | **79.91** | 79.99 |

**Notes.**
Best results are shown in bold.

number of nodes in the subgraph increases when k equals 5, and the performance of inductive link prediction does not improve significantly. On the contrary, it decreases on some datasets. This suggests that although an increase in the number of hops introduces more information about nodes, not all of the introduced nodes are necessarily useful for link prediction. Some of them may be noisy nodes, which do not benefit the performance of relation prediction.

• Ablation study

We conduct ablation studies on FB15K237_v2 to validate the importance of relation semantics for subgraph extraction in SASILP. Table 4 shows the results of the ablation

**Table 3** Inductive link prediction performance on FB15K237.

| Datasets | Model | MRR | Hits@1 | Hits@5 | Hits@10 |
|---|---|---|---|---|---|
| FB15K237_v1 | Neural-Lp | 46.13 | 40.21 | 52.08 | 52.92 |
| | DRUM | 47.55 | 42.71 | 51.46 | 52.92 |
| | RuleN | 45.97 | 41.16 | 49.51 | 49.76 |
| | GraiL | 48.56 | 40.00 | 58.54 | 64.15 |
| | SNRI | 48.82 | 38.05 | 59.02 | 73.90 |
| | LCILP | 47.15 | 38.04 | 56.1 | 58.29 |
| | RMPI | **51.65** | **44.15** | 59.12 | 65.07 |
| | SASILP | 49.23 | 41.02 | **59.93** | **65.43** |
| FB15K237_v2 | Neural-Lp | 51.85 | 45.68 | 58.06 | 58.94 |
| | DRUM | 52.78 | 47.49 | 57.93 | 58.73 |
| | RuleN | 69.08 | **62.13** | 76.78 | 77.82 |
| | GraiL | 62.54 | 52.20 | 75.21 | 81.80 |
| | SNRI | 64.72 | 53.34 | 77.62 | **85.77** |
| | LCILP | 56.68 | 47.07 | 67.78 | 72.91 |
| | RMPI | 65.42 | 55.88 | 77.45 | 81.57 |
| | SASILP | **71.02** | 60.02 | **78.43** | 82.54 |
| FB15K237_v3 | Neural-Lp | 48.70 | 44.09 | 52.46 | 52.90 |
| | DRUM | 49.64 | 45.84 | 52.63 | 52.90 |
| | RuleN | **73.68** | **65.95** | 83.12 | 87.69 |
| | GraiL | 70.35 | 60.25 | 82.36 | 82.83 |
| | SNRI | 65.37 | 54.91 | 78.03 | 85.66 |
| | LCILP | 50.74 | 42.81 | 60.4 | 63.7 |
| | RMPI | 65.5 | 56.67 | 75.75 | 81.53 |
| | SASILP | 71.26 | 65.73 | **84.64** | **88.76** |
| FB15K237_v4 | Neural-Lp | 49.54 | 44.12 | 54.81 | 55.88 |
| | DRUM | 50.43 | 45.53 | 54.88 | 55.88 |
| | RuleN | 74.19 | **67.21** | 82.27 | 85.60 |
| | GraiL | 70.60 | 60.99 | 82.62 | 89.29 |
| | SNRI | 62.75 | 51.05 | 76.79 | 85.50 |
| | LCILP | 65.34 | 57.02 | 76.45 | 80.84 |
| | RMPI | 66.62 | 55.79 | 80.41 | 87.21 |
| | SASILP | **75.67** | 66.04 | **83.45** | **89.97** |

**Notes.**
Best results are shown in bold.

study. After removing the relation semantic score in subgraph extraction, the MRR value reduces by 8.34%, and the Hits@5 value reduces by 18.36%. This result demonstrates the effectiveness of relation semantics in inductive link prediction.

● Case Study

To analyze the subgraph under different subgraph extraction strategies, we selected two nodes (07s9rl0, 06 × 77 g) in FB15K237_v2 as seed nodes. The subgraphs extracted using GraiL contain 1,218 nodes, the value of Hits@10 is 52.2%, While SASILP extracted using SASILP contains 11 nodes, the value of Hits@10 is 60.02%. Figure 7 shows the semantics of the extracting subgraph, where the purple circle represents the two seed nodes and the

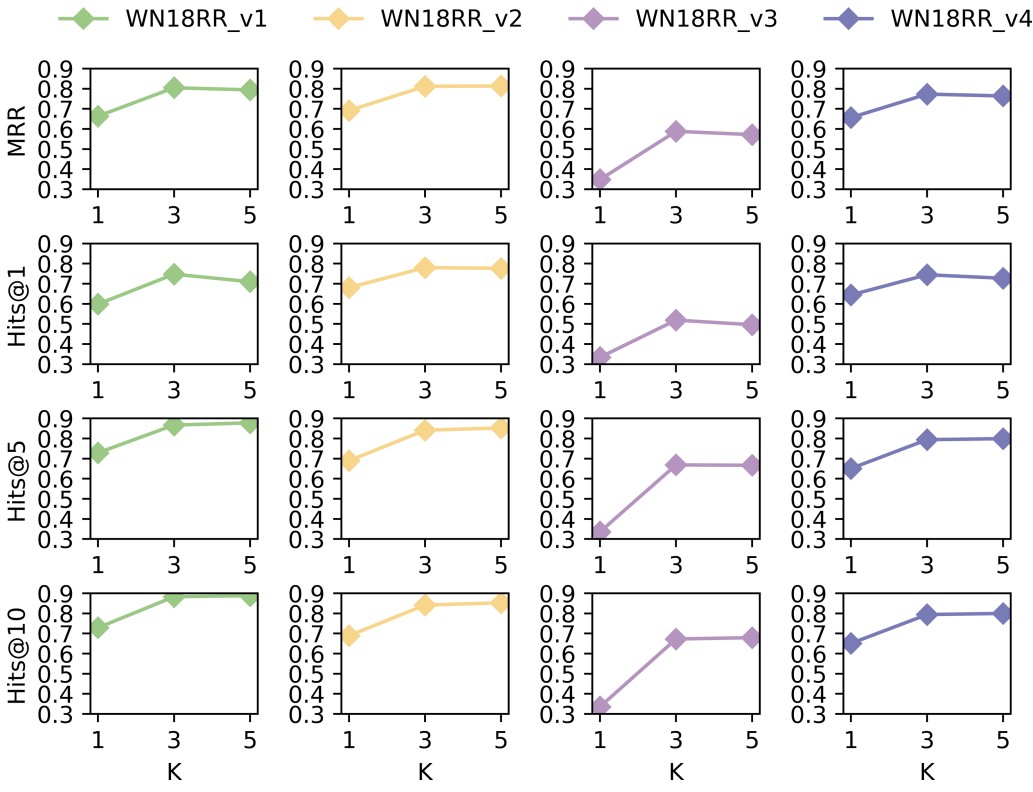

**Figure 6** Performance of SASILP on k-hop.

**Table 4** Ablation results of MRR and Hits@5 on FB15K237_v2.

| Method | MRR | Hits @ 5 |
|---|---|---|
| SASILP w/o realtion semantic | 62.68 | 60.07 |
| SASILP | 71.02 | 78.43 |

yellow dotted line represents the relation to be predicted. The number of subgraph nodes decreased sequentially, but it did not influence the link prediction performance. It may be caused that the nodes in the subgraph have a close semantic correlation with the missing relation, and it also demonstrates the effectiveness of our subgraph extraction method.

## CONCLUSIONS AND FUTURE WORK

We propose a novel model SASILP for inductive link prediction in knowledge graphs. The model focuses on the subgraph extraction and node initialization methods. The subgraph is constructed by nodes that are selected according to the importance score. The node is initialized by integrating structure and relational semantic information. The experimental

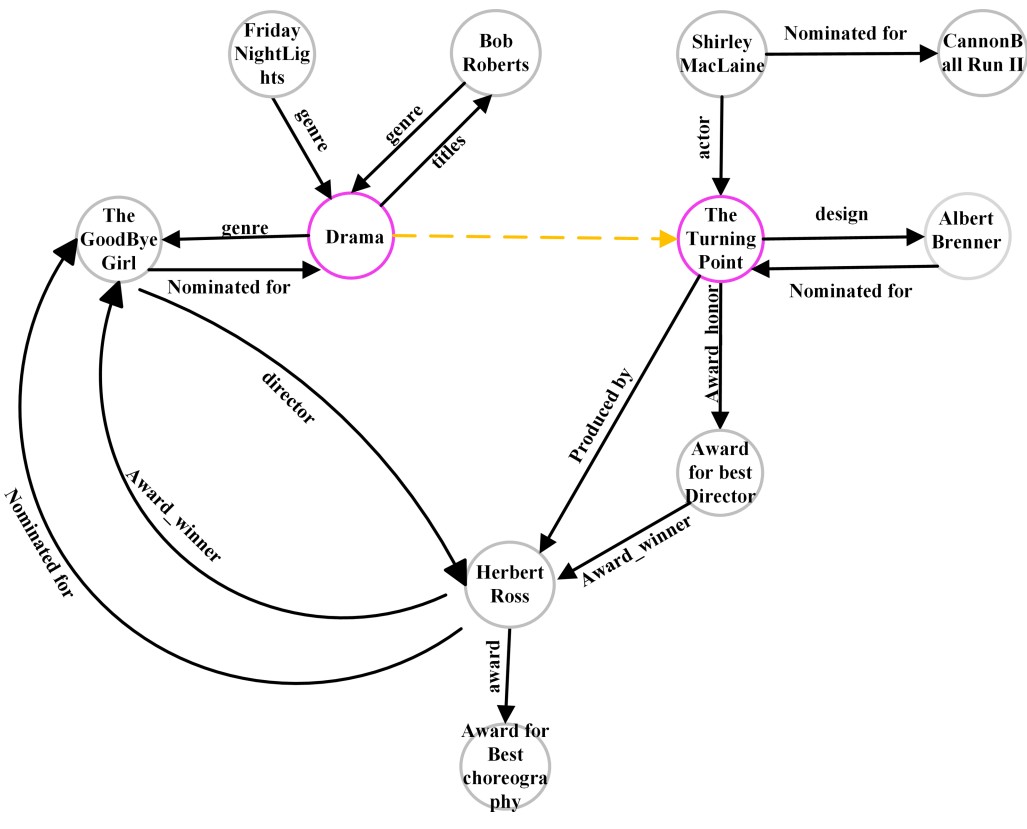

**Figure 7** A subgraph extracting from FB15K237.

results demonstrate that SASILP outperforms the baseline models on the WN18RR and FB15K237 datasets.

Most existing models only apply to scenarios where there are new entities in the target graph. They cannot handle the real-world knowledge graphs where new entities accompany new relations. Therefore, we plan to investigate the link prediction scenarios that contain both new entities and relations in the knowledge graph.

### Funding
This work was supported by the National Natural Science Foundation of China (No. 62172433). The funders had no role in study design, data collection and analysis, decision to publish, or preparation of the manuscript.

### Grant Disclosures
The following grant information was disclosed by the authors:
the National Natural Science Foundation of China: No. 62172433.

## Competing Interests

The authors declare there are no competing interests.

## Author Contributions

- Hongbo Liu conceived and designed the experiments, performed the experiments, analyzed the data, performed the computation work, authored or reviewed drafts of the article, and approved the final draft.
- Jicang Lu analyzed the data, prepared figures and/or tables, authored or reviewed drafts of the article, and approved the final draft.
- Tianzhi Zhang performed the computation work, prepared figures and/or tables, and approved the final draft.
- Xuemei Hou performed the computation work, prepared figures and/or tables, and approved the final draft.
- Peng An performed the experiments, authored or reviewed drafts of the article, and approved the final draft.

## Data Availability

The dataset is available at GitHub:

https://github.com/kkteru/grail (owned by Komal K. Teru, komal.teru@mail.mcgill.ca, McGill University.)

## Supplemental Information

Supplemental information for this article can be found online at http://dx.doi.org/10.7717/peerj-cs.2324#supplemental-information.

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
