# Peer review of "Relation semantic fusion in subgraph for inductive link prediction in knowledge graphs"

_PeerJ Computer Science, doi:10.7717/peerj-cs.2324_

## Round 0.1 · original submission · Major Revisions

Thank you for submitting your manuscript for review. Based on the feedback from our reviewers, we have identified several areas that require significant revisions before your article can be considered for publication. Please note that addressing all the points mentioned below does not guarantee acceptance in the second round, but it will certainly improve the overall quality and clarity of your manuscript.

Summary of Required Revisions:
Firstly, the clarity and consistency of the manuscript need to be improved. The title should be rewritten for better clarity. Ensure consistency in notations and references throughout the article, such as maintaining the same font for all characters in equations and consistent figure references. Figures 1 and 3 need to be clearer, with better explanations of symbols and improved appearance.

The abstract should be enriched to explain how the model uses structural and semantic information. In the introduction, highlight the novelty of the SASILP model compared to existing methods and provide specific, quantitative performance comparisons with state-of-the-art models, rather than generic statements like "significant performance improvements."

The research questions and objectives need to be clearly defined to make them easier to understand. The new components added to the model should be presented more clearly, ensuring they align with the overall structure of the article. It is essential to highlight the major enhancements over baseline models more clearly and ensure proper citation of related works, particularly the LCILP model (Locality-aware subgraphs for inductive link prediction in knowledge graphs, Mohamed et al., 2023).

In the Related Work section, include references to critical works on Embedding-based Link Prediction, such as “Knowledge graph representation learning with simplifying hierarchical feature propagation, IP&M,” “Text-enhanced knowledge graph representation learning with local structure, IP&M,” and “Multi-scale Dynamic Convolutional Network for Knowledge Graph Embedding, TKDE.”

The Method section requires more detailed explanations, particularly regarding the importance of relational semantics and structure. Enrich the description of the subgraph extraction process in Figure 2 and consider revising or deleting Figure 4 if it is not discussed in the text.

In the Experimental Design and Results section, include more baseline models for comparison and ensure these are up-to-date with recent state-of-the-art methods. Add ablation results to verify the validity of model components.

Lastly, conduct a thorough grammatical check to correct any mistakes throughout the paper.

We hope that you find the reviewers' comments constructive and that they help you in improving your manuscript. Please address all the points mentioned above comprehensively in your revised submission.

We look forward to receiving your revised manuscript.

Reviewer 1 ·

Basic reporting

In this study, the authors developed an inductive link prediction model that learns logical rules to predict unseen entities. The core feature of this method is its enhanced capability to extract data from graphs. However, several issues need to be addressed to improve the readability of the study:
- The title should be rewritten for clarity.
- Notations and references should be consistent (e.g., "Re" in "LeakyReLU" in Equation 2 should have the same font as other characters; figure references are inconsistently written as "figure 3" and "Figure3").
- Figure 3 is confusing because it does not explain the meaning of the solid and dashed arrow symbols.

Experimental design

The research questions in the study are not clearly defined, making it difficult to understand the specific objectives the authors aim to address. Additionally, the new components added to the model should be presented more clearly in accordance with the overall structure of the article.

Validity of the findings

- The performance of the model has improved, but not significantly compared to the LCILP model (Locality-aware subgraphs for inductive link prediction in knowledge graphs, Mohamed et al., 2023), which this method is based on without proper citation.
- The study lacks novelty.
- The major enhancements made over baseline models are not clearly highlighted.

Reviewer 2 ·

Basic reporting

Research summary: An inductive link prediction model based on subgraph extraction is proposed. The key idea is to use relational semantics to extract subgraphs and initialize nodes, to construct subgraphs of target relationships. The authors also conducted many experiments on real data sets to show good results in the link prediction task.

For this article, I have the following concerns:

1. When describing the SASILP model, the abstract section can more specifically explain how the model uses structural and semantic information. For example, you can briefly describe the architecture or critical components of the model.
2. In the introduction, it is best to highlight the novelty of the SASILP model compared to existing methods. For example, the model has a unique advantage in dealing with new entities or relationships in the inductive task and should be clearly stated. In the main contribution, try to describe the specific performance of the model on these datasets, rather than just mention "significant performance improvements", and provide quantitative comparisons with existing state-of-the-art models to demonstrate the extent of the improvement. In addition, figure 1 is not clear enough, and the appearance is not good, so the size and spacing need to be adjusted.
3. In Related Work, some critical works on Embedding-based Link Prediction are missing:
“Knowledge graph representation learning with simplifying hierarchical feature propagation, IP&M”
“Text-enhanced knowledge graph representation learning with local structure, IP&M”
“Multi-scale Dynamic Convolutional Network for Knowledge Graph Embedding, TKDE”
4. Models and methods for inductive link prediction can be discussed in relevant literature.
5. In Method, Figure 2 (a) does not explain and distinguish the importance of relational semantics and structure, the subgraph extraction part is too simple, and the overall process needs to be enriched. The author should have done a better job of redrawing Figure 2, refining more details, and keeping the frame diagram compact and orderly. (b) Figure 4 is not mentioned in the article, so please consider deleting it.
6. In experiments, (a) there are too few baseline models for comparison. (b) The state-of-the-art baseline method is older. Authors are encouraged to compare their methods with the latest state-of-the-art methods. (c) It is recommended that ablation results be added to verify the validity of the model components.
7. This paper contains several grammatical mistakes. I suggest a thorough check. I do not recommend publishing this paper in its present form.

Experimental design

no comment

Validity of the findings

no comment

Additional comments

no comment

---

## Round 0.2 · accepted · Accept

Dear authors,

I am pleased to inform you that your manuscript titled "Relation Semantic Fusion in Subgraph for Inductive Link Prediction in Knowledge Graphs" has been accepted for publication in [Journal Name]. The reviewers have evaluated the revisions, and I am happy to confirm that all necessary changes have been satisfactorily addressed.

Thank you for your diligent work in revising the manuscript. We are excited to include your research in our upcoming issue and believe it will make a significant contribution to the field.

Best regards,

Reviewer 2 ·

Basic reporting

The authors have addressed all my issues.

Experimental design

no comment

Validity of the findings

no comment

Additional comments

no comment